# A Social Business Model of Early Intervention and Rehabilitation for People with Disability in Rural Bangladesh

**DOI:** 10.3390/brainsci12020264

**Published:** 2022-02-14

**Authors:** Mahmudul Hassan Al Imam, Manik Chandra Das, Israt Jahan, Mohammad Muhit, Delwar Akbar, Nadia Badawi, Gulam Khandaker

**Affiliations:** 1CSF Global, Dhaka 1213, Bangladesh; physiomahmud@yahoo.com (M.H.A.I.); manikchandradas10@gmail.com (M.C.D.); arda.jahan89@gmail.com (I.J.); mmuhit@hotmail.com (M.M.); 2Asian Institute of Disability and Development (AIDD), University of South Asia, Dhaka 1213, Bangladesh; 3School of Health, Medical and Applied Sciences, Central Queensland University, Rockhampton, QLD 4701, Australia; 4Central Queensland Public Health Unit, Central Queensland Hospital and Health Service, Rockhampton, QLD 4700, Australia; 5School of Business and Law, Central Queensland University, Rockhampton, QLD 4701, Australia; d.akbar@cqu.edu.au; 6Cerebral Palsy Alliance, Sydney Medical School, The University of Sydney, Camperdown, NSW 2050, Australia; nadia.badawi@health.nsw.gov.au; 7Discipline of Child and Adolescent Health, Sydney Medical School, The University of Sydney, Sydney, NSW 2050, Australia

**Keywords:** early intervention, rehabilitation, community-based rehabilitation, disability, cerebral palsy, social business model, sustainability

## Abstract

Background: Despite the high burden of childhood disability in low-and middle-income countries (LMICs), the opportunity for early intervention and rehabilitation is very limited. Studies have found that community-based rehabilitation service is effective for children with cerebral palsy (CP); however, such services are not readily available in LMICs, and services run by non-profit organisations on external funding are often not sustainable. In this study, we report the lesson learnt in establishing a social business model of early intervention and rehabilitation services for children with CP and adults with disabilities in a rural subdistrict of Bangladesh. Methods: Case study of a rural early intervention and rehabilitation centre (i.e., the model centre) implemented between May 2018 and September 2019. An economic evaluation incorporating gross margin analysis along with descriptive statistics was performed to assess the social business potentials of the model centre. Results: The establishment of this model centre cost ~5955 USD with an average monthly running cost of ~994 USD. During the 17 months study period, 7038 therapy sessions (average eight sessions per patient) were offered to 862 patients with musculoskeletal and neurological disorders. The most common clinical presentations were low back pain (35.6%; *n* = 307). Six percent (*n* = 52) of the attendees were children with CP (mean (SD) age 6.3 (4.0) years; 35.7% (*n* = 19) were female), who received 1392 sessions, on average 27 sessions per child. The centre reached the break-even point at the 13th month and remained profitable for the next 4 months of the study period. An average session fee of 2.2 USD resulted in a gross margin of -1458 USD and 1940 USD in 2018 and 2019, respectively. Revenue to cost ratios for the 2 years were 0.27:1 and 0.51:1 while average rates of return were −41.4% and 10.1%, respectively. Sensitivity analysis revealed that session numbers including 5000, 6000, 7000, 8000, 9000, and 10,000 were required to break even at the session fees of 3.0, 2.50, 2.0, 2.0, 1.5, and 1.5 USD, respectively. Conclusion: Our social business model of an early intervention and rehabilitation service provides evidence of enhancing access to services for children with CP as well as adults with disabilities while ensuring the sustainability of the services in rural Bangladesh.

## 1. Introduction

The prevalence and severity of cerebral palsy (CP) are strikingly higher in low- and middle-income countries (LMICs) compared to high-income countries [1,2]. On the other hand, the opportunity to access rehabilitation services in LMICs is very limited because of a lack of trained rehabilitation professionals [3]. The findings from the Bangladesh Cerebral Palsy Register (BCPR), an ongoing population-based surveillance for children with CP based in Bangladesh, revealed that there are an estimated ~234,000 children with CP in Bangladesh and the age of CP diagnosis is significantly delayed [1]. Moreover, almost half of the children with CP registered with the BCPR do not have access to rehabilitation services [4]. A recent situation analysis study claimed that there are less than 10 physiotherapists for every 1 million population, and the majority (93.8%) of them are located in urban cities in Bangladesh [5]. These findings underscore the need for an innovative community-based service delivery model to improve access to rehabilitation for these geographically disadvantaged children with CP in Bangladesh.

Little is known about the existing community-based rehabilitation (CBR) service delivery models for children with CP in LMICs. Dambi et al. [6] compared CBR with the institute-based rehabilitation (IBR) for children with CP in Zimbabwe. Compared to IBR, CBR was associated with greater improvement in compliance, motor function, and satisfaction [6]. Employing the ‘Getting to Know CP’ training manual, Zuurmond et al. [7] evaluated a participatory parent training program for children with CP in Ghana. The authors observed significant improvements in parents’ quality of life, knowledge about CP, feeding practices, and perception about the physical and emotional health of their child [7]. With a randomised controlled trial (RCT) design, the outcomes of a peer-delivered parents training are being assessed in India [8]. As part of the trial, parents of children with disabilities were identified from local communities and trained on goal-directed training, motor and cognitive enrichment training (using learning toys), and parent educational modules [8]. Karim et al. [9] examined the effects of a community-based early intervention and rehabilitation program for children with CP in rural Bangladesh. As part of the program, group therapy, community follow-up, and primary caregivers training are provided for 6 months free of cost [9]. The authors observed significant improvements in motor function and communication of children with CP who attended the early intervention program compared with their non-attendee peers [9]. To assess the outcome of an integrated livelihood and CBR program, an RCT is being conducted among children with CP from ultra-poor families in rural Bangladesh [10]. The trial included three study arms: integrated livelihood and CBR arm, CBR alone arm, and control arm [10]. As part of the trial, children of the integrated livelihood and CBR arm received weekly CBR sessions and livelihood support (e.g., livestock, sewing machines), while the CBR alone arm received weekly CBR sessions and the control arm received basic education on early intervention and rehabilitation [10]. However, the sustainability of such CBR services, in the long run, remains mostly unknown.

Although a number of CBR models have been found to be effective in improving the health and wellbeing of children with CP and their parents in LMICs; however, the majority of them are dependent on external funds. To date, no study has examined how a community-based service delivery model for children with CP can become self-sustainable in an LMIC context. To ensure sustainability, a paradigm shift from a conventional charity-based model to a social business model has been recommended [11]. However, the potential of a rehabilitation service centre for children with CP as a social business model has not been examined yet. This study aimed to assess the outcome of a social business model of early intervention and rehabilitation services for children with CP and adults with disabilities in a rural subdistrict of Bangladesh.

The organisation of this article is as follows: The first section describes the background of the study; the second section illustrates the theoretical framework of the study; the third section explains the methodologies adopted; the fourth section narrates the findings of the study; the fifth section states the discussions, limitations, and recommendations; and the sixth section outlines the conclusions.

## 2. Theoretical Framework

Noble laureate professor Muhammad Yunus stated that the traditional social service providers or non-government organisations could become self-reliant by transforming their services into social businesses [11]. Social business has been defined as ‘a self-sustaining company that sells goods or services and repays its owners’ investments, but whose primary purpose is to serve society’ [11]. In a social business model, the traditional profit-maximising businesses and not-for-profit organisations are brought under a single umbrella with an aim to act as a change agent for the world (Figure 1) [11]. According to Nishith Desai [12], the sluggish rate of human development in spite of high gross domestic product (GDP) and insufficient participation of government in ensuring basic public, social, and economic services to its relatively disadvantaged citizens, were the underpinning factors behind the development of the social business model.

The social business model emphasises a new development approach with its own set of opportunities and limitations. The key concept of social business is that the profit equation should not only consider the financial returns but also it should take into account welfare-enhancing outcomes to bring about the positive social impact [11,13]. Social business ventures can turn poor communities from “aid” beneficiaries into active market stakeholders, and short-term development projects into financially viable sustainable programs [14]. The social business approach drives entrepreneurs to shift from business-as-usual, to develop business innovation and create local markets [15]. In Bangladesh, the social business initiatives of the Grameen Bank have contributed to poverty reduction [16,17] and provided access to safe water [18] and nutritious foods [18,19]. However, this literature review did not find any empirical study reporting a social business model providing services for persons with disabilities.

Yunus et al. [11] recommended that the social business model should consider three strategic moves: challenging conventional knowledge, developing partnerships, and ongoing experimentations. There is a knowledge gap regarding the potential of a disability service as a social business model. Existing disability services are either for-profit or not-for-profit ventures in Bangladesh [5]. However, previous studies indicate that a health service can also be a social business model if the right approach is adopted [11]. We, therefore, hypothesise that early intervention and rehabilitation services in rural Bangladesh can be a self-sustainable social business model. The implementing organisation CSF Global (www.csf-global.org, accessed on 19 January 2022) has a strong track record in childhood research and early intervention and rehabilitation services in LMICs including Bangladesh. Therefore, we did not require a partner organisation. Since 2015, CSF Global has established nine CBR centres for children with CP in six rural sub-districts of Bangladesh. Over time, CSF Global experimented with this CBR service model in different locations and with different service types and fee structures. This helped to develop a road map for rolling out the social business concept: a self-sustaining program that sells services for people with disabilities and reimburses its owners for their investments, but whose primary purpose is to serve society by improving access to services for the disadvantaged communities.

## 3. Materials and Methods

### 3.1. Study Design, Setting, and Participants

As part of the BCPR [1], the authors developed and piloted a social business model of early intervention and rehabilitation centre between May 2018 and September 2019. Following a case study design, we report our experiences from the pilot project, which was implemented at Nabinagar subdistrict in the Chittagong division of Bangladesh. The subdistrict consists of 353.7 sq km area and 420,383 population [20].

The model community-based early intervention and rehabilitation centre was established in May 2018. The model centre was staffed by a physiotherapist, a community therapist, and a centre manager, and equipped with basic physiotherapy equipment (e.g., treadmill, traction unit, ultrasound therapy unit, multiple mode electrical stimulators, assistive devices, parallel bars, wedges). Following the establishment of the centre, consultation meetings with relevant stakeholders (e.g., health workers, non-government organisations, religious leaders) and mobile health camps were organised in order to raise awareness about the services being provided at the centre as well as to develop referral networks. The centre followed evidence-based clinical guidelines (e.g., activity-focused therapy, Goal-Directed Training and Constraint-Induced Movement Therapy) and offered group therapy and individual therapy sessions for children with CP free of cost. Additionally, clinical assessment and 1 hour of physiotherapy sessions were provided for adult clients with musculoskeletal conditions (low back pain, neck pain, and knee pain) and neurological disorders (e.g., stroke and facial palsy) in exchange for a nominal fee (i.e., total cost of ~2 USD per session). However, the session charge was subsidised up to 100% in case of financial constraints of adult clients. Revenue generated from the adult patient services (e.g., musculoskeletal pain and post-stroke rehabilitation) was used to provide subsidies for the early intervention and rehabilitation program for children with CP.

### 3.2. Data Management, Analysis, and Ethical Considerations

A structured data collection template was used to collect the data on patient’s socio-demographic and clinical characteristics (e.g., age, sex, education, chief complaint, and diagnosis) and services (e.g., number of patients, income, expenditures, and subsidies). The data were collected both in a paper-based template and an Excel spreadsheet, and the spreadsheets were shared with the study team on a daily basis. To assess the business potential of the centre, an economic evaluation was completed, incorporating the gross margin analysis and sensitivity of the gross margin analysis. The use of gross margin analysis in assessing business models has been reported in earlier studies [21,22]. The following formula was used to calculate gross margin: gross margin = total revenue - total variable cost. Here, total revenue stands for total income and total variable costs include personnel costs, facility costs, utility costs, and miscellaneous costs of the centre in a financial year (July 2018–June 2019). Additionally, descriptive statistics on the patient data were performed to assess the number and types of patients attending the centre. Written informed consent was obtained from the clients or primary caregivers. This pilot study was approved by the Human Research Ethics Committee (HREC) of the Asian Institute of Disability and Development (AIDD) (Reference: Southasia-hrec-2018-2-04).

## 4. Results

The findings of the pilot study showed a constant growth of the model centre in terms of the number of sessions provided and revenues generated. The establishment of this centre cost 5955 USD with an average monthly running cost of 994 USD (Table 1).

During the study period, 862 patients (7038 sessions, average 8 sessions per patient) with musculoskeletal and neurological disorders (mean (standard deviation (SD)) age 45.2 (28.7), 47.6% (*n* = 410) female, and 33.4% secondary education level completed) received services at the model centre. Most common clinical presentations were low back pain (35.6%; *n* = 307), knee pain (17.4%; *n* = 150), and neck pain (10.1%; *n* = 87). Among the attendees, 6.0% (*n* = 52) were children with CP (mean (SD) age 6.3 (4.0) years and 35.7% (*n* = 19) were female), who received 1392 sessions, on average 27 sessions per child. Spastic quadriplegia (69.2%; *n* = 36) was the major type of CP followed by spastic diplegia (15.4%; *n* = 8) and mixed CP (7.7%; *n* = 4) (Table 2).

Out of 7038 sessions, 1958 (27.8%) sessions worth 3014 USD were subsidised for ultra-poor people who could not afford the service otherwise. During 17 months of operation, the centre reached the break-even point at the 13th month and remained profitable for the next 4 months of the study period (Figure 2).

Between July 2018 and June 2019 (financial year of Bangladesh), the social business model offered 5203 therapy sessions incurring 19333 USD. The variable cost accounted for 66.1% of the total costs. Based on the sessions and costs of 2018, an estimated USD 13,379 was needed to offer 6769 sessions in 2019. These estimates are based on the assumptions that no fixed costs other than depreciation costs are required and there will be a 30% increment in the session number in 2019. There was a 76% growth in the session number between May 2018 (279) and June 2019 (491). An average session fee of 2.2 USD resulted in a gross margin of −1458 USD and 1940 USD in 2018 and 2019, respectively. The revenue to cost ratios for the two years were 0.27:1 and 0.51:1, while the average rates of return were −41.4% and 10.1%, respectively (Table 3).

Table 4 presents sensitivity analysis on the social business model of early intervention and rehabilitation centre to observe the effect of price fluctuation and session number on gross margin. Session numbers including 5000, 6000, 7000, 8000, 9000, and 10,000 were required to break even at the session fees of 3.0, 2.50, 2.0, 2.0, 1.5, and 1.5 USD, respectively.

After ensuring the sustainability of the centre, the franchise of the model centre was handed over to a local team to empower and encourage the local community to take over the ownership and provide services for children with CP. As of October 2021, the centre has been rendering full-fledged services without requiring subsidies for its running costs. The BCPR team still provides ongoing technical support to maintain the service standards of the centre.

## 5. Discussion

To the best of our knowledge, this is the first study assessing the potential of a rehabilitation service centre from a social business perspective in LMICs. The findings of this pilot study suggest that an early intervention and rehabilitation centre can grow into a self-reliant service model within 13 months of operation even after offering free services for children with CP and subsidised services for adults with disabilities.

Recent studies on the social business model in public health initiatives have shown promising outcomes. For instance, the collaboration between Grameen Bank (a leading microfinance institution in Bangladesh) and Danone (a leading healthy food company in the world) has ensured affordable and accessible dairy food products to fulfill the nutritional needs of children in Bangladesh [11]. Similarly, the collaborative effort from Grameen Bank and Veolia Water (one of the world’s leading water service providers) has facilitated the development of a simplified surface-water treatment system to provide affordable access to drinking water with using a pre-paid payment system for rural populations [11]. Another collaborative project titled Bangladesh Sprinkles Program was developed by BRAC (the leading non-government organisation in Bangladesh), Renata limited (a leading pharmaceutical company in Bangladesh), and the Global Alliance for Improved Nutrition (GAIN) to produce and distribute Pushtikona (a sachet containing 15 essential minerals and vitamins), with an aim to prevent anaemia, growth retardation, and gastrointestinal infections [23]. With the use of the pharmacy network of Renata limited [24] and ~97,000 health workers of BRAC [25], the nutritional intervention could reach the poorest of the poor in exchange for a nominal fee (~3 cents USD) in Bangladesh [23]. A recent study on Pushtikona reported that it is a highly cost-effective method for distributing micronutrients to a large number of children in Bangladesh [26]. However, our literature search did not reveal any such study focusing on rehabilitation of people with disabilities.

The importance of developing a social business model of early intervention and rehabilitation service is also reiterated in earlier studies from LMICs. Al Imam et al. [27] observed a substantially delayed diagnosis and a very poor rehabilitation service uptake among children with CP in LMICs including Bangladesh, Nepal, Indonesia, and Ghana. The authors recommended that a community-based innovative service delivery model is essential in order to enhance rehabilitation service utilisation among these vulnerable children [27]. The need for a community-based affordable service delivery model has also been emphasised in a recent study conducted on children with CP in Bangladesh [4]. Ensuring adequate access to services (e.g., physiotherapy in our business model) is essential to facilitate early diagnosis and early initiation of rehabilitation services [28]. Our experience from this self-sustainable model centre suggests that rehabilitation services could be a successful business model while making an extensive social impact (e.g., improved access and utilisation of services for children and adults with disabilities).

Our social business model has several economical and policy level implications. The provision of free and subsidised early intervention services at our model centre eventually benefitted the country economically. Studies have shown that early initiation of intervention and rehabilitation can prevent functional loss and disabilities [29,30], and thus, it can help to reduce disability-adjusted life years [31]. The prevention of disability also helps the health system by reducing the length of hospital stays, hospital burden, and associated costs [32]. This knowledge has important implications to reform existing policies and encourage public–private partnerships to facilitate such social business ventures. An affordable and locally available service delivery model is essential to meet the 2030 agenda for Sustainable Development Goals of ensuring equal access to rehabilitation services among this vulnerable population.

Despite our best effort, this study inherited some limitations. Firstly, our study is mainly based on a single centre from one subdistrict only and, therefore, the findings of this study cannot be generalised. Nevertheless, the Nabinagar subdistrict represents rural and semi-urban Bangladesh in terms of demographics and other indicators [23]. Secondly, the study did not report the cost-effectiveness in terms of expenditures incurred to obtain certain health outcomes of children with CP. However, this study aimed to present the potential of a rehabilitation service delivery model for children with CP and adults with disabilities as a social business venture. Thirdly, we could not report the proportion of children with CP and/or adults with disabilities living in the study site who had access to this service. Future health system research on such a social business model is needed to develop early intervention and rehabilitation services for children with CP and other neurodevelopmental disabilities in low-resource settings.

## 6. Conclusions

This study assessed the outcome of a social business model of early intervention and rehabilitation services for children with CP and adults with disabilities in a rural subdistrict of Bangladesh. The findings of the study suggest that an early intervention and rehabilitation centre can become a social business venture, reaching a break-even point within a year of operation. Our social business model provides evidence of enhancing access to services for children with CP as well as adults with disabilities, while ensuring the sustainability of the services in rural Bangladesh. This social business model of community-based services could be an effective strategy to facilitate early diagnosis and improve access and utilisation of rehabilitation services for children with CP in LMICs. This evidence has significant policy implications in terms of scaling up such a business model at the national level in Bangladesh and other LMICs.

## Figures and Tables

**Figure 1 brainsci-12-00264-f001:**
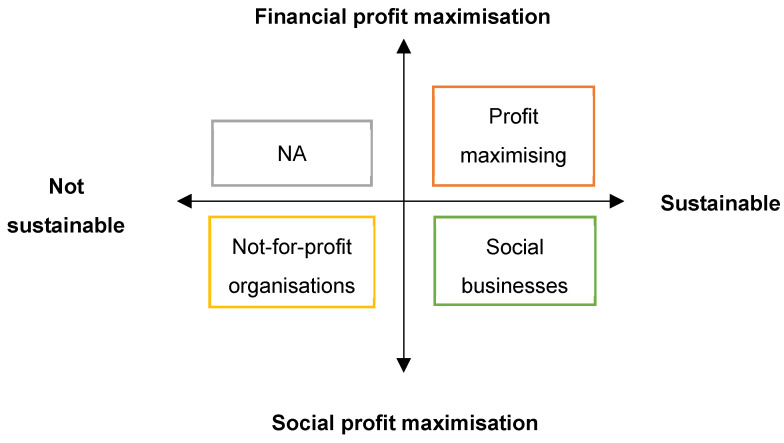
Social businesses vs. profit-maximising businesses and not-for-profit organisations adapted from Yunus et al. [11].

**Figure 2 brainsci-12-00264-f002:**
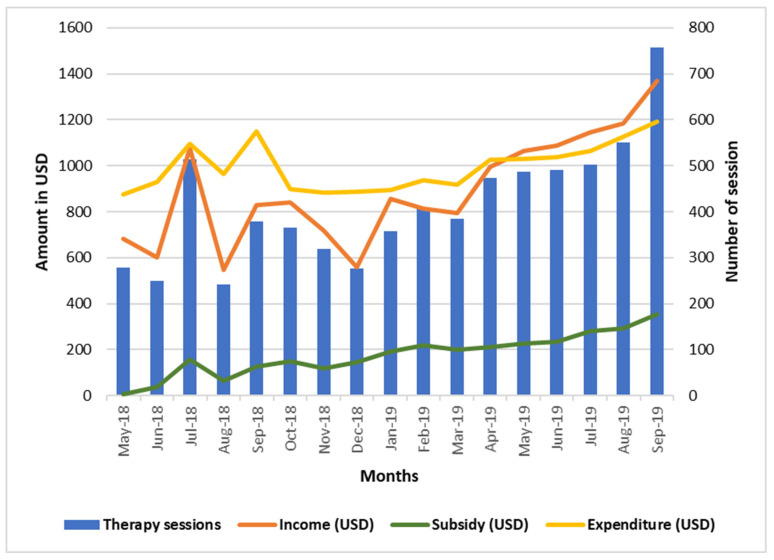
Distribution of therapy sessions, income, expenditure, and subsidy in the social business model of early intervention and rehabilitation centre.

**Table 1 brainsci-12-00264-t001:** The establishment and running costs of the social business model of early intervention and rehabilitation centre.

Budget Heads	Budget Details	Amount in USD
Establishment costs (one off)	Therapy equipment (e.g., treadmill, traction unit, multiple mode electrical stimulator, assistive devices)	2231
	Furniture (e.g., beds, tables, chairs), electronics (e.g., computer, ceiling fans, lights) and stationaries (e.g., register books, staplers, pen stands)	3035
	Printing items (e.g., signboard, forms, invoices, leaflets, festoon)	332
	Repair and renovation (e.g., painting, curtain set up)	357
	Sub-total	5955
Variable cost or running cost per month	Personnel costs (e.g., salary of physiotherapist, community therapist)	702
	Facility cost (e.g., rent, maintenance)	143
	Utility costs (electricity bill, water bill, generator bill)	113
	Miscellaneous (e.g., photocopy, cleaning items, transport, courier etc.)	36
	Sub-total	994

**Table 2 brainsci-12-00264-t002:** Socio-demographic and clinical characteristics of the clients attending the model centre.

Characteristics	Categories	*n* (%)
Age	mean (SD)	45.2 (28.7)
Sex	Male	452 (52.4%)
	Female	410 (47.6%)
Education	Illiterate	132 (24.2%)
	Primary completed	164 (30.1%)
	Secondary completed	182 (33.4%)
	Higher secondary and above	67 (12.3%)
Clinical presentations	Low back pain	307 (35.6%)
	Knee pain	150 (17.4%)
	Neck pain	87 (10.1%)
	Shoulder pain	72 (8.4%)
	Elbow pain	64 (7.4%)
	Cerebral palsy	52 (6.0%)
	Stroke	49 (5.7%)
	Wrist pain	37 (4.3%)
	Facial palsy	19 (2.2%)
	Others (e.g., post-fracture complications, hip joint pain, multiple joint pain)	25 (2.9%)
Type of Cerebral Palsy	Quadriplegia	36 (69.2%)
	Diplegia	8 (15.4%)
	Mixed	4 (7.7%)
	Dystonia	2 (3.8%)
	Athetosis	2 (3.8)

**Table 3 brainsci-12-00264-t003:** Fixed cost, variable cost, revenue, and gross margin for the social business model.

Item	Cost (USD/Session) 2018–2019	Estimated Cost (USD/Session) 2019–2020	Total Costs (USD) in July 2018–June 2019	Estimated Total Costs (USD) in July 2019–June 2020	Average Cost in USD/Session
Fixed cost					
Centre set up	1.1		5954		1.1
Depreciation costs for equipment and furniture	0.1	0.1	595.4	595.4	0.1
Total fixed cost	1.3	0.1	6549.4	595.4	1.3
Variable cost					
Personnel costs (e.g., staff salary)	1.66	1.66	9029	9029	1.7
Facility costs (e.g., rent, maintenance)	0.34	0.34	1839	1839	0.3
Utility costs (electricity bill, water bill, generator bill)	0.27	0.27	1453	1453	0.3
Miscellaneous (e.g., photocopy, cleaning items, transport, courier, etc.)	0.08	0.08	463	463	0.1
Total variable cost	2.3	2.3	12,784.0	12,784.0	2.3
Total expenses	3.6	2.5	19,333.4	13,379.4	3.6
Revenue Segment	Unit Price	Total (2018)	Total (2019) *	per session (2018)	per session (2019)
Average Income/session in 2018–2019	2.2	11,326	14,723.7	2.2	2.2
Session in 2018–2019		5203	6763.9		
Total revenue (in USD)		11,326	14,724	906.1	669.3
Gross margin (in USD)		−1458	1940	−116.6	88.2
Total profit (in USD)		−8007	1344	−640.6	61.1
Revenue: Cost ratio		0.3	0.5		
Average rate of return		−41.4	10.1		

* Considering 30% increment.

**Table 4 brainsci-12-00264-t004:** Sensitivity analysis of the average gross margin from 2 years of the social business model.

Fee per Session (USD)	Session per Year					
	5000	6000	7000	8000	9000	10,000
0.5	−10,284.0	−9784.0	−9284.0	−8784.0	−8284.0	−7784.0
1.0	−7784.0	−6784.0	−5784.0	−4784.0	−3784.0	−2784.0
1.5	−5284.0	−3784.0	−2284.0	−784.0	716.0	2216.0
2.0	−2784.0	−784.0	1216.0	3216.0	5216.0	7216.0
2.5	−284.0	2216.0	4716.0	7216.0	9716.0	12,216.0
3.0	2216.0	5216.0	8216.0	11,216.0	14,216.0	17,216.0

## Data Availability

The data presented in this study are available on request from the corresponding author. The research data contain potentially sensitive and identifying patient information and, therefore, the data are not publicly available.

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
