# Peer review of "A Social Business Model of Early Intervention and Rehabilitation for People with Disability in Rural Bangladesh"

_brainsci, 2022, doi:10.3390/brainsci12020264_

Round 1

Reviewer 1 Report

This manuscript addresses an important issue about the disability-specific social business model providing early intervention and rehabilitation services in a rural sub-district of Bangladesh, and authors concluded that social business model of early intervention and rehabilitation service provides evidence of enhancing access to services for children with CP as well as adults with disabilities while ensuring the sustainability of the services in rural Bangladesh .

I think although this paper has economic aspects, it fully conveys the importance of rehabilitation for CP children in n low-and middle-income 
countries . If there is no problem with the economic methodology, it is considered acceptable.

Author Response

Thank you for the positive feedback.

Reviewer 2 Report

Thank you for giving me the possibility to review this paper. I hope that the authors find my comments productive and that they help them to improve their research work.

In this paper the authors present disability-specific social business model providing early intervention and rehabilitation services in a rural sub-district of Bangladesh.

The proposed Keywords are correct and correspond to the research topic.

In the Introduction the authors should indicate the year and cite correctly when Nishith Desai is mentioned. In addition, the authors should state the objective and the main motivations of the research as well as put the topic of the study in context with relevant data and graphs. Authors are asked to include this information in the penultimate paragraph. The structure of the paper should be properly explained in the last paragraph. Authors are asked to include in the last paragraph of this section the structure of the paper. With regard to the structure of the paper, authors are asked to reorganise the information in the large blocks of 1)Introduction, 1) Theoretical framework or Literature review, 3) Materials and methods or Methodology, 4) Data analysis or Results, 5) Discussion, 6) Conclusions.

It is necessary to include in the article a section that refers to the Literature Review/Theoretical framework and where the main studies and research related to the research topic that have led the authors to carry out their research are included. In addition, this section should include the hypothesis of the research, i.e., the problem or gap that has been detected and about which we have made a prediction that later, in the methodology, we have been able to corroborate or not. In addition, authors are asked to bear in mind that the hypotheses should be based on the literature review.

In Materials and Methods the authors correctly describe how they have designed the study and the most important aspects of the study. However, it would be advisable if the authors can refer to previous studies or research that have used the same methodology in order to justify their validity.

Regarding the Results section, its presentation could be improved, and it is recommended that the authors include Tables 3 and 4 of the Discussion section in the Results section, which is where they correspond.

The Discussion section is correct, explaining the limitations of the study and proposing new and future lines of research. In addition, this section also presents the results obtained with previous research in the same field of study.  In the same section, authors are asked to cite Imam et al. correctly, including the year of publication.

The Conclusion section, although correct, is scarce. Authors are asked to consider that this section is intended to clarify the objectives of the research, to state the aim of the research and to show what the authors are demonstrating with their research. Authors are asked to improve this section.

The references are correct but few. Authors are recommended to improve the number of references based on the above comments.

Author Response

31 Jan 2022

Professor Dr. Stephen D. Meriney

Editor-in-Chief

Brain Sciences

RE: Manuscript ID brainsci-1535585

Title: A Social Business Model of Early Intervention and Rehabilitation for People with Disability in Rural Bangladesh

We would like to thank the reviewers for their constructive feedback and helpful comments on our manuscript titled ‘A Social Business Model of Early Intervention and Rehabilitation for People with Disability in Rural Bangladesh’. Please see below our point-by-point responses to the reviewers’ comments.

Reviewer-1

Comment-1: This manuscript addresses an important issue about the disability-specific social business model providing early intervention and rehabilitation services in a rural sub-district of Bangladesh, and authors concluded that social business model of early intervention and rehabilitation service provides evidence of enhancing access to services for children with CP as well as adults with disabilities while ensuring the sustainability of the services in rural Bangladesh.

I think although this paper has economic aspects, it fully conveys the importance of rehabilitation for CP children in n low-and middle-income countries. If there is no problem with the economic methodology, it is considered acceptable.

Our response: Thank you for the positive feedback.

Reviewer-2:

Comment-1:

In this paper the authors present disability-specific social business model providing early intervention and rehabilitation services in a rural sub-district of Bangladesh. The proposed Keywords are correct and correspond to the research topic.

Our response: Thank you for the feedback.

Comment-2: In the Introduction the authors should indicate the year and cite correctly when Nishith Desai is mentioned. In addition, the authors should state the objective and the main motivations of the research as well as put the topic of the study in context with relevant data and graphs. Authors are asked to include this information in the penultimate paragraph. The structure of the paper should be properly explained in the last paragraph. Authors are asked to include in the last paragraph of this section the structure of the paper. With regard to the structure of the paper, authors are asked to reorganise the information in the large blocks of 1)Introduction, 1)Theoretical framework or Literature review, 3) Materials and methods or Methodology, 4) Data analysis or Results, 5) Discussion, 6) Conclusions.

Our response: Thank you for the constructive suggestions. We have now added corrected the citation of the mentioned sentence (please see line 123).

As suggested, we have added the objective and the rationale of the study in the penultimate paragraph. Please see below the revised paragraph:

“Although a number of CBR models have been found to be effective in improving the health and wellbeing of children with CP and their parents in LMICs; however, the majority of them are dependent on external funds. To date, no study has examined how a community-based service delivery model for children with CP can become self-sustainable in an LMIC context. To ensure sustainability, a paradigm shift from a conventional charity-based model to a social business model is necessary (11). However, the potential of a rehabilitation service centre for children with CP as a social business model has not been examined yet. This study aimed to assess the outcome of a social business model of early intervention and rehabilitation services for children with CP and adults with disabilities in a rural subdistrict of Bangladesh.”

A paragraph explaining the structure of the paper has now been added at the end of the background section. Please see below the newly added paragraph:

“The organisation of this article is as follows: the first section describes the background of the study, the second section illustrates the theoretical framework of the study, the third section explains the methodologies adopted, the fourth section narrates the findings of the study, the fifth section states the discussions, limitations, and recommendations and the sixth section outlines the conclusions.” 

In addition, we have also included a relevant reference as background information (lines 76-82) to support the research gap.

Comment-3: It is necessary to include in the article a section that refers to the Literature Review/Theoretical framework and where the main studies and research related to the research topic that have led the authors to carry out their research are included. In addition, this section should include the hypothesis of the research, i.e., the problem or gap that has been detected and about which we have made a prediction that later, in the methodology, we have been able to corroborate or not. In addition, authors are asked to bear in mind that the hypotheses should be based on the literature review.

Our response: Thank you for the important suggestion about the theoretical framework. We have now added a new section entitled ‘Theoretical Framework’. In this section, we have described the relevant literature, determined the hypothesis/knowledge gaps in terms of the existing social business model, and outlined the rationale for conducting the study. Please see the below the Theoretical Framework section:

“2. Theoretical Framework

            Noble laureate professor Muhammad Yunus stated that the traditional social service providers or non-government organisations could become self-reliant by transforming their services into social businesses (11). Social business has been defined as ‘a self-sustaining company that sells goods or services and repays its owners' investments, but whose primary purpose is to serve society’ (11). In a social business model, the traditional profit-maximising businesses and not-for-profit organisations are brought under a single umbrella with an aim to act as a change agent for the world (Figure-1) (11). According to Nishith Desai (12), the sluggish rate of human development in spite of high gross domestic product (GDP) and insufficient participation of government in ensuring basic public, social, and economic services to its relatively disadvantaged citizens were the underpinning factors behind the development of the social business model.

Figure 1. Social businesses vs. profit maximising businesses and not-for-profit organisations adapted from Yunus et al. (11).

            Social business model emphasises a new development approach with its own set of opportunities and limitations. The key concept of social business is that the profit equation should not only consider the financial returns but also it should take into account welfare-enhancing outcomes to bring about the positive social impact (11, 13). Social business ventures can turn poor communities from “aid” beneficiaries into active market stakeholders, and short-term development projects into financially viable sustainable programs (14). Social business approach drives entrepreneurs to shift from business-as-usual, develop business innovation, and create local markets (15). In Bangladesh, the social business initiatives of the Grameen Bank have contributed to poverty reduction (16, 17) and provided access to safe water (18) and nutritious foods (18, 19). However, this literature review did not find any empirical study reporting a social business model providing services for persons with disabilities.

Yunus et al. (11) recommended that the social business model should consider three strategic moves: challenging conventional knowledge, developing partnerships, and ongoing experimentations. There is a knowledge gap regarding the potential of a disability service as a social business model. Existing disability services are either for-profit or not-for-profit ventures in Bangladesh (5). However, previous studies indicate that a health service can also be a social business model if the right approach is adopted (11). We, therefore, hypothesise that early intervention and rehabilitation services in rural Bangladesh can be a self-sustainable social business model. The implementing organisation CSF Global (www.csf-global.org) has a strong track record in childhood research and early intervention and rehabilitation services in LMICs including Bangladesh. Therefore, we did not require a partner organisation. Since 2015, CSF Global has established nine CBR centres for children with CP in six rural sub-districts of Bangladesh. Over time, CSF Global experimented with this CBR service model in different locations and with different service types and fee structures. This helped to develop a road map for rolling out the social business concept: a self-sustaining program that sells services for people with disabilities and repays its owners for their investments, but whose primary purpose is to serve society by improving access to services for the disadvantaged communities.”

Comment-4: In Materials and Methods the authors correctly describe how they have designed the study and the most important aspects of the study. However, it would be advisable if the authors can refer to previous studies or research that have used the same methodology in order to justify their validity.

Our response: Thank you for the valuable suggestions. We have now cited previous studies that followed a similar methodology. Please see the below new sentence and citations added to the methodology section:

“The use of gross margin analysis in assessing business models has been reported in earlier studies (21, 22).”

Comment-5: Regarding the Results section, its presentation could be improved, and it is recommended that the authors include Tables 3 and 4 of the Discussion section in the Results section, which is where they correspond.

Our response: Thank you for the suggestion to improve presentations. We have revised the results section incorporating all tables within the results section. We also request the editing team to keep both Table 3 and Table 4 in the Results section.

Comment-6: The Discussion section is correct, explaining the limitations of the study and proposing new and future lines of research. In addition, this section also presents the results obtained with previous research in the same field of study. In the same section, authors are asked to cite Imam et al.correctly, including the year of publication.

Our response: Thank you for the positive feedback. As suggested, we have corrected the citation of Imam et al. (Please see line 281).

Comment-7: The Conclusion section, although correct, is scarce. Authors are asked to consider that this section is intended to clarify the objectives of the research, to state the aim of the research and to show what the authors are demonstrating with their research. Authors are asked to improve this section.

Our response: Thank you for the valuable feedback. We have now revised the conclusion section restating the objective of the study and describing our key findings. Please see the revised Conclusion section as follows:

“This study assessed the outcome of a social business model of early intervention and rehabilitation services for children with CP and adults with disabilities in a rural subdistrict of Bangladesh. The findings of the study suggest that an early intervention and rehabilitation centre can become a social business venture reaching a break-even point within a year of operation. Our social business model provides evidence of enhancing access to services for children with CP as well as adults with disabilities while ensuring the sustainability of the services in rural Bangladesh. This social business model of community-based services could be an effective strategy to facilitate early diagnosis and improve access and utilisation of rehabilitation services for children with CP in the majority world population. This evidence has significant policy implications in terms of scaling up such a business model at the national level in Bangladesh and other LMICs.”

Comment-8: The references are correct but few. Authors are recommended to improve the number of references based on the above comments.

Our response: Thank you for the valuable feedback. We have now added 09 new references. The revised total number of references is 32.

Yours sincerely,

Professor Gulam Khandaker

This manuscript is a resubmission of an earlier submission. The following is a list of the peer review reports and author responses from that submission.